# Experience and lessons learned from multi-modal internet-based recruitment of U.S. Vietnamese into research

**Milkie Vu**[1]*, **Victoria N. Huynh**[2], **Robert A. Bednarczyk**[3,4], **Cam Escoffery**[1,4], **Danny Ta**[5], **Tien T. Nguyen**[6], **Carla J. Berg**[7,8]

**1** Department of Behavioral, Social, Health Education Sciences, Rollins School of Public Health, Emory University, Atlanta, Georgia, United States of America, **2** Emory College of Arts & Sciences, Emory University, Atlanta, Georgia, United States of America, **3** Hubert Department of Global Health, Rollins School of Public Health, Emory University, Atlanta, Georgia, United States of America, **4** Winship Cancer Institute, Emory University, Atlanta, Georgia, United States of America, **5** Nell Hodgson Woodruff School of Nursing, Emory University, Atlanta, Georgia, United States of America, **6** Mount Holyoke College, South Hadley, Massachusetts, United States of America, **7** Department of Prevention and Community Health, Milken Institute of Public Health, George Washington University, Washington, District of Columbia, United States of America, **8** George Washington Cancer Center, George Washington University, Washington, District of Columbia, United States of America

* milkie.vu@emory.edu

**Data Availability Statement:** The data that support the findings of this study are available upon reasonable request and with the approval of Emory

## Abstract

### Background

Asian-Americans are one of the most understudied racial/ethnic minority populations. To increase representation of Asian subgroups, researchers have traditionally relied on data collection at community venues and events. However, the COVID-19 pandemic has created serious challenges for in-person data collection. In this case study, we describe multi-modal strategies for online recruitment of U.S. Vietnamese parents, compare response rates and participant characteristics among strategies, and discuss lessons learned.

### Methods

We recruited 408 participants from community-based organizations (CBOs) (n = 68), Facebook groups (n = 97), listservs (n = 4), personal network (n = 42), and snowball sampling (n = 197). Using chi-square tests and one-way analyses of variance, we compared participants recruited through different strategies regarding sociodemographic characteristics, acculturation-related characteristics, and mobile health usage.

### Results

The overall response rate was 71.8% (range: 51.5% for Vietnamese CBOs to 86.6% for Facebook groups). Significant differences exist for all sociodemographic and almost all acculturation-related characteristics among recruitment strategies. Notably, CBO-recruited participants were the oldest, had lived in the U.S. for the longest duration, and had the lowest Vietnamese language ability. We found some similarities between Facebook-recruited participants and those referred by Facebook-recruited participants. Mobile health usage

University Institutional Review Board (IRB 00111688). The data are not publicly available as it contains information that could compromise the privacy of research participants. In particular, given that participants are Vietnamese parents of adolescents (a small, specific population), and that the dataset contains their zip code, making the data available can risk the possibility of participants being identified. Please see more information here: http://www.irb.emory.edu/documents/phi_identifiers.pdf Please contact the lead author, MV, as well as the Institutional Review Board at Emory University, with any request for data access (milkie.vu@emory.edu and IRB@emory.edu).

**Funding:** This work is supported by the American Psychological Foundation 2019 Visionary Grant and the American Association for Cancer Education 2019 Grant in Research, Education, Advocacy, and Direct Service (READS), the Grants-in-Aid program from the Society for the Psychological Study of Social Issues, the Professional Development Support Fund at Emory University, and the Healthcare Innovation Program Student-Initiated Project Grant at the Georgia Clinical & Translational Science Alliance (CTSA). The Georgia CTSA is supported by the National Center for Advancing Translational Sciences of the National Institutes of Health under Award Number UL1TR002378. Our data collection receives support from the Center for AIDS Research at Emory University (P30AI050409). Ms. Vu is supported by the US National Cancer Institute (5F31CA243220-02), a 2020-2021 PEO Scholar Award, and the 2020-2021 Student Fellowship in Patient Engagement from the Society of Public Health Education. Dr. Berg is supported by the US National Cancer Institute (R01CA215155-01A1; R01CA179422-01; R01CA239178-01A1), the US Fogarty International Center/National Institutes of Health (1R01TW010664-01), and the US National Institute on Environmental Health Science/Fogarty International Center (D43ES030927-01). Dr. Bednarczyk is supported in part by the US National Cancer Institute (1R37CA234119-01). Open-access publication support was made possible in part by the Research Reimbursement Award from grant 3P30CA076292 (Geographic Management of Cancer Health Disparities Program (GMaP) Region 2) funded by the National Cancer Institute.

**Competing interests:** The authors have declared that no competing interests exist.

was high and did not vary based on recruitment strategies. Challenges included encountering fraudulent responses (e.g., non-Vietnamese). Perceived benefits and trust appeared to facilitate recruitment.

## Conclusions

Facebook and snowball sampling may be feasible strategies to recruit U.S. Vietnamese. Findings suggest the potential for mobile-based research implementation. Perceived benefits and trust could encourage participation and may be related to cultural ties. Attention should be paid to recruitment with CBOs and handling fraudulent responses.

## Introduction

Asians are the fastest growing racial/ethnic groups in the U.S. [1], with a population of more than 18 million as of 2019 (6% of the total population) [2], which is projected to grow to 41 million (9% of the total population) by 2050 [3]. Despite their growing numbers, Asian-Americans remain one of the most understudied racial/ethnic minority groups [4–7]. For example, studies including Asian-American participants totaled 0.01% of MEDLINE/PubMed articles published between 1966 and 2000 [4] and 0.17% of the NIH-funded clinical research budget between 1992 and 2018 [5]. Moreover, when Asian-American participants are included, research typically aggregates Asian subgroups into one single category [6–8] instead of providing data for separate subgroups. This trend masks important subgroup differences, as Asian-Americans are extraordinarily heterogeneous, comprising of people from over 30 countries and speaking over 100 languages/dialects, with diverse socioeconomic status, religious and cultural backgrounds, immigration histories, and patterns of health services utilization [8–11].

While in recent years, more large-scale or population-based datasets and registries (e.g., the National Health and Nutrition Examination Survey or the National Health Interview Survey) have incorporated questions assessing separate Asian subgroups in their survey, due to small sample sizes, research using these datasets has rarely reported disaggregated subgroup outcomes [6, 7]. Moreover and critically, with a few notable exceptions (e.g., the National Latino and Asian American Study, the California Health Interview Survey), large-scale or population-based surveys often are not available in Asian languages, thus excluding the participation of those with lower English proficiency [6, 7]. According to the 2019 American Community Survey, 74% and 31% of the Asian population in the U.S. reported that they spoke a language other than English and that they spoke English less than "very well," respectively [2]. The literature has demonstrated several differences between Asians with and without lower English proficiency. In particular, compared to Asians with higher English proficiency, Asians with lower English proficiency have higher psychological distress [12, 13], higher unmet healthcare needs [14], poorer quality of life and health [15, 16], more limited access to care [14] and are less likely to adhere to screening guidelines [17] or receive and use health services [13, 18].

To overcome these issues, many researchers interested in Asian-American health have relied on the use of community-engaged research [19, 20]. Typically, researchers partner with community-based organizations (CBOs) serving Asian populations to collect data from clients using CBO services or attending community venues and events [7, 21–25]. This approach has allowed for the administration of surveys in diverse languages and the collection of adequate sample sizes from different Asian subgroups, thereby providing important representation and understanding of Asian-American health behaviors and outcomes.

The ongoing COVID-19 pandemic in the U.S., however, represents a serious challenge to research employing in-person modes of data collection. A growing emphasis is being placed on transitioning to online data collection [26, 27]. To successfully execute these methods in Asian communities, an understanding of the success and challenges of different recruitment strategies is critical. Yet, only a handful of published studies to date have described the process of online recruitment and data collection in Asian-Americans populations in detail [28–35].

Existing literature found that, in general, Asian participants recruited from Internet sources were more likely to be younger, U.S.-born, fluent in English, and more educated than Asian participants recruited from non-Internet sources [28, 29]. Researchers also highlighted several issues in the online recruitment of Asian participants, notably difficulties in establishing trust and transparency between researchers and participants as well as participants' concerns about the security of information transmitted through the Internet [30]. In addition, challenges in creating rapport with "gatekeepers" (e.g., webmasters, Facebook group moderators, directors of Asian community groups) were also a barrier to recruitment [31, 32]. On the other hand, having a culturally matched research team (e.g., same ethnicity or strong cultural ties) [31, 33] and clear communication and outline of research procedures [31] were facilitators to recruitment.

The primary goal of this paper is to contribute to the body of literature on strategies for online data collection efforts in Asian communities by presenting a case study of a research project with U.S. Vietnamese (i.e., those living in the U.S. and identifying as Vietnamese). U.S. Vietnamese represent the fourth largest Asian-American subgroup, with a population of more than 1.8 million as of 2019 [2]. According to the 2019 American Community Survey, U.S. Vietnamese have lower English proficiency, median household income, and education and are more likely to live in poverty compared to other Asians [2]. These disadvantages reflect potential barriers to health services utilization and health research participation, thus highlighting the need to focus on this specific subgroup (rather than Asians as an aggregate).

Indeed, a body of literature has documented lower utilization of various health services among U.S. Vietnamese when compared to other major Asian subgroups. For example, compared to Chinese and Korean populations in the U.S., fewer U.S. Vietnamese have a personal doctor as a main healthcare provider [36], have ever been tested for Hepatitis B [37], or have ever had colorectal screening [38]. Moreover, a higher proportion of U.S. Vietnamese women have never had a Pap smear compared to Chinese and Cambodian women [39]. Additionally, fewer U.S. Vietnamese women on average have ever sought mental health services compared to Chinese women or Filipino women in the U.S [40].

The data presented in this paper are from a study investigating multilevel factors influencing U.S. Vietnamese parents' HPV vaccine initiation for their adolescent children. U.S. Vietnamese have higher cervical cancer incidence rates than other racial/ethnic groups [41–44]. A solution to reduce cervical cancer burden in this population is HPV vaccination. Unfortunately, our understanding of HPV vaccination among U.S. Vietnamese is limited, partly because large national probability surveys on HPV vaccine uptake in the U.S. typically do not supply separate statistics for U.S. Vietnamese. For example, the National Immunization Survey–Teen and the Behavioral Risk Factor Surveillance System, both of which provide estimates on HPV vaccine uptake, aggregate "Vietnamese" under the category of "Asians" and do not provide disaggregated Asian subgroup data in public-use datasets [45–47]. The Health Information National Trends Survey (HINTS), which examines knowledge of the HPV vaccine, reports separate data for Vietnamese in some cycles but not all; the sample sizes for Vietnamese in the public-use datasets are also relatively small (~1%) which can hinder meaningful statistical modeling [48]. Importantly, none of these surveys includes Vietnamese language versions of their questionnaires, which can be a major obstacle to research participation for U.

S. Vietnamese with low English proficiency. In the context of research with HPV vaccination, this issue is particularly problematic, given that U.S. Vietnamese women with lower English proficiency also had lower HPV vaccine uptake [49]. Consequently, excluding those with low English proficiency could bias estimates of HPV vaccine uptake and lead to an incomplete understanding of barriers underlying HPV vaccination in this population.

In this study, we describe multiple strategies for online recruitment of U.S. Vietnamese parents and compare response rates among strategies. We also compare several characteristics (e.g., sociodemographic, acculturation, and mobile health usage) of participants drawn from different strategies. Moreover, we discuss lessons learned from our experiences conducting online recruitment, including handling fraudulent responses and how perceived benefits and trust can encourage participation.

## Materials and methods

### Eligibility criteria and research procedure

Eligibility criteria included: 1) self-identified as Vietnamese; 2) having lived in the U.S. for at least 12 months; 3) able to read either Vietnamese or English; and 4) having at least one child aged 9 to 18 and living in the same household with the child. Only one parent per household was allowed to participate. Participants first completed an online eligibility screener (administered via SurveyGizmo). Those eligible were directed to a webpage with a study description and a consent form. Consenting individuals indicated consent via a link and were directed to the online survey (via SurveyGizmo). The survey took approximately 60 minutes to complete. We limited duplicate responses by permitting only one response per IP address. Survey participants were compensated with a $30 Amazon gift card. We also selected a subset of 32 participants from those who completed the survey to participate in semi-structured interviews to further explore findings from the survey. We obtained electronic/online consent prior to the interview for each participant. Interviewed participants were compensated with a $60 Amazon gift card. Participants could choose to do the survey and interview in either Vietnamese or English. The Institutional Review Board at Emory University approved of this study.

### Recruitment procedures: Survey

Survey recruitment took place virtually from April to December 2020. We utilized several venues for survey recruitment: community-based organizations, Facebook groups, listservs, the personal network of the first author, and snowball sampling [50]. Convenience sampling was used for all recruitment methods.

**Community-based organizations.** We contacted CBOs serving Asians and/or Vietnamese populations in the U.S., introduced the research study and eligibility criteria, and asked the CBOs to send information about our research to their members. Additionally, we contacted Vietnamese Students Associations (VSA) at colleges and universities and requested VSA students to have their parents contact us if their parents met the eligibility criteria and were interested.

**Fig 1** details the recruitment process with CBOs.

In total, we contacted 115 Vietnamese-serving CBOs, 91 Asian-serving CBOs, and 114 VSAs. Among those, 40 Vietnamese-serving CBOs, 20 Asian-serving CBOs, and 13 VSAs agreed to disseminate information about the study to their members. Six Vietnamese-serving CBOs and one Asian-serving CBO declined to participate in the project; reasons for nonparticipation included lack of members in the targeted demographics (n = 2), no email addresses of members on file (n = 1), insufficient staff capacity due to COVID-19 (n = 2), and insufficient technological knowledge among members to navigate online surveys (n = 2). In total, 51

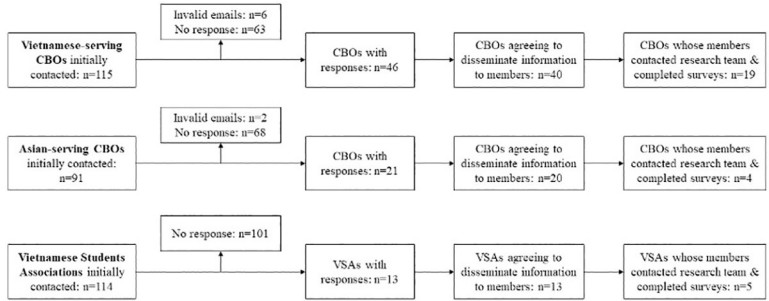

**Fig 1. Recruitment process with community-based organizations.**

participants from 19 Vietnamese-serving CBOs, nine participants from four Asian-serving CBOs, and eight participants from five VSAs completed the surveys.

**Facebook groups.** Through a search on Facebook with terms such as "Vietnamese in the U.S.", we identified Facebook groups that operated in Vietnamese and focused on topics that might be of interests to U.S. Vietnamese parents of adolescents (e.g., general discussion of life in the U.S., parenting advice for those with children in the U.S., immigration and visa applications, etc.). We posted information about the research study and eligibility criteria in 12 Facebook groups, which ranged in membership from 50 to approximately 58,000. Ninety-seven participants from these 12 Facebook groups completed the surveys.

**Listservs.** We posted information about the research study and eligibility criteria to two listservs, one focusing on Vietnamese Studies and the other on Vietnamese-related events in California. Four participants (two from each listserv) contacted the study team and completed the surveys.

**Personal network.** The first author is a U.S. Vietnamese, fully fluent in Vietnamese and English, and has several years of experience partnering with Vietnamese and Asian-serving CBOs in different U.S. regions and conducting research on Vietnamese health. She disseminated information about the research study and eligibility criteria to her personal network via social media and emails. She requested those who were eligible and interested to contact the study team. She also asked others in her personal network to refer eligible and interested acquaintances to the study team. With this method, 42 participants completed the surveys.

**Snowball sampling.** Using snowball sampling, we asked participants who completed the surveys to refer their eligible and interested acquaintances to the study team. A total of 197 participants completed the surveys through snowball sampling recruitment, in which two were referred from previous participants recruited through Vietnamese-serving CBOs, two referred from previous participants recruited through VSAs, 77 referred from those recruited through Facebook groups, and 116 referred from those recruited through the first author's personal network.

**Survey measurements.** The survey was available in Vietnamese and English. For the surveys, we used the Brislin's back-translation method [51], an iterative translation process involving an independent translation of survey questions into Vietnamese and back-translation into English by two different translators, and then reviewed by the first author (who is fully fluent in both languages) and by approximately 10 Vietnamese native speakers to ensure comprehensibility.

For **sociodemographic characteristics**, we assessed a participant's age, sex, highest education level, combined household income, and child's country of birth.

For **acculturation-related characteristics**, we assessed a participant's percentage of life in the U.S. and their ability to understand medical information in English. In addition, we asked

about participants' zip codes and used data from the 2019 American Community Survey [52] to construct two variables capturing zip code-level percentage of Asians and percentage of Vietnamese.

We also used the Asian American Multidimensional Acculturation Scale (15 items for each culture) [53] to separately assess cultural identity (6 items), cultural knowledge (3 items), language (4 items), and food consumption (2 items) for Vietnamese culture and American culture. Examples of questions included: "How much do you interact and associate with [Vietnamese people]/[typical American people]?"; "How much do you actually practice the traditions and keep the holidays of [Vietnamese culture]/[mainstream American culture]?"; "How well do you speak [Vietnamese]/[English]?"; and "How often do you actually eat [Vietnamese food]/[the food of mainstream American culture]?" Each item was scored on a 6-point Likert scale (0 –Not very much to 5 –Very much). A higher subscale score, derived as an average across subscale items, indicated higher cultural identity, cultural knowledge, language ability, or food consumption.

For **mobile health usage**, we used items from the Health Information National Trends Survey 5 Cycle 2 [54] to assess participants' daily use of a home computer or mobile device to access the Internet, past 12-month use of electronic devices to look for health-related information for themselves or their children, and past 12-month use of emails or the Internet to communicate with a doctor. Given the critical role of digital technologies in facilitating data collection during the COVID-19 pandemic, these data can indicate whether access to the Internet (and, consequently, ability to participate in online data collection) differs between participants from different recruitment methods and potentially explain differences in response rates. In addition, they can also provide information on the possible receptiveness of participants to research and interventions leveraging mobile health technologies.

## Recruitment procedure: Interviews

Between November 2020 and February 2021, we invited a subset of participants who had already completed the survey to participate in in-depth, semi-structured qualitative interviews. We purposively sampled participants and stratified the sample of interviewees by their adolescent child's sex (female versus male) and HPV vaccination status. For selected potential interviewees, we sent each of them an email with information about the interviews and a consent form. Depending on participants' preferences, interviews were conducted in either Vietnamese or English and via telephone or the Zoom platform. Of 38 invited participants, 32 agreed to participate in the interviews (84% response rate). Among the 32 interviewed participants, 7 (21.9%) were recruited through CBOs, 1 (3.1%) through listservs, 10 (31.3%) through Facebook groups, 1 (3.1%) through personal network, 8 (25.0%) referred from those recruited through Facebook groups, and 5 (15.6%) referred from those recruited through the first author's personal network.

**Interview questions.** Interview questions explored participants' sources of information about the HPV vaccine. While these questions were not designed specifically to inquire participants' thoughts about the research project, several participants brought up the research project as a resource that motivated them to learn more about the HPV vaccine.

## Data analysis

We used the American Association for Public Opinion Research's (AAPOR) response rate calculator version 4.1 to determine response rates [55]. Chi-square tests and one-way analyses of variance were used to compare differences in sociodemographic characteristics, acculturation-related characteristics, and mobile health usage among recruitment methods. To further

understand significant differences between recruitment methods, the Tukey test was used for post-hoc analyses of differences in pairs of means, while a Tukey-type procedure was used for post-hoc analyses of differences in pairs of proportions [56]. Analyses were conducted in SAS 9.4 and Stata 15.1 and alpha levels were set at 0.05. Qualitative interview transcripts were uploaded to MAXQDA 2020. We used a hybrid approach of qualitative thematic analysis, which incorporated both 1) a deductive *a priori* template of codes and themes included in the interview guide and relevant literature and 2) a data-driven inductive approach that identified new emergent themes [57]. Two coders (MV and DT) coded all transcripts independently. Coding results were compared and any discrepancies were resolved through discussion.

## Results

### Response rates

Table 1 summarizes the number of completed surveys as well as response rates with different venue.

The overall response rate was 71.8% (range: 51.5% for Vietnamese CBOs to 86.6% for Facebook groups). Fig 2 shows the recruitment flowchart for the entire sample.

Fig 3 shows the percentages of participants in the survey sample from different recruitment methods.

### Sample characteristics

**Sociodemographic characteristics.** We found differences in all sociodemographic characteristics among recruitment methods (Table 2).

On average, participants recruited through CBOs were the oldest (45 years). Post-hoc analyses show that, compared to those recruited through CBOs, a significantly higher proportion of those recruited through Facebook groups reported that they had a Bachelor's degree or higher (66.2% versus 87.6%) and that their child was born outside of the U.S. (22.1% versus

**Table 1. Responses and response rates among recruitment methods.**

| Type of recruitment method | Completed responses (I) | Partial responses (P)* | Eligible but refused participation or did not start (R) | Unknown eligibility (NC)** | Response rate |
|---|---|---|---|---|---|
| Community-based organizations | | | | | |
| Vietnamese-serving community-based organizations | 51 | 2 | 6 | 40 | 51.5 |
| Asian-serving community-based organizations | 9 | 0 | 1 | 1 | 81.8 |
| Vietnamese Students Association | 8 | 2 | 1 | 3 | 57.1 |
| Facebook groups | 97 | 4 | 4 | 7 | 86.6 |
| Listservs | 4 | 1 | 2 | 0 | 57.1 |
| Personal network | 42 | 5 | 1 | 9 | 73.7 |
| Snowball sampling | | | | | |
| Through community-based organizations (Vietnamese CBO and VSA) | 4 | 0 | 0 | 1 | 80.0 |
| Through Facebook | 77 | 3 | 4 | 18 | 75.5 |
| Through personal network | 116 | 10 | 4 | 28 | 73.4 |

Using the American Association for Public Opinion Research's (AAPOR) response rate calculator version 4.1, the response rate for each recruitment method was calculated as $I/[(I + P) + (R + NC)]$

* Column does not show 1 individual referred from those recruited through listservs (i.e. snowball sampling through listservs)

** Column does not show 2 individuals referred from those recruited through listservs (i.e. snowball sampling through listservs).

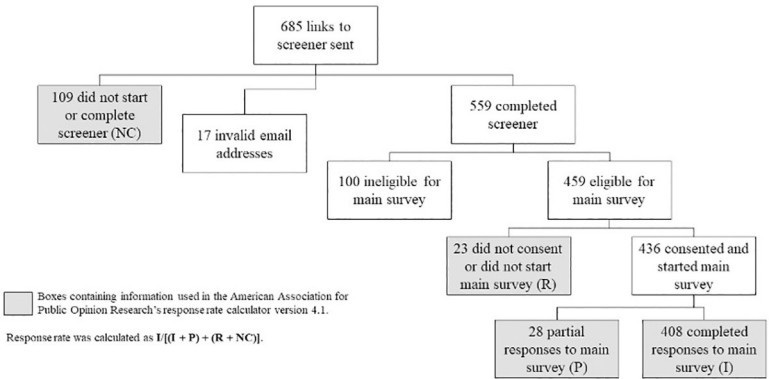

**Fig 2. Recruitment flowchart for the entire sample.**

70.1%). The distribution of education appeared similar between those recruited through Facebook groups, snowball sampling via Facebook groups, and the first author's personal network.

**Acculturation-related characteristics.** Except for a Vietnamese acculturation subscale (food consumption), we found significant differences in all acculturation-related characteristics among recruitment methods (**Table 3**).

On average, Facebook groups-recruited participants and those referred by them had lived in the U.S. for the shortest durations, had the highest scores on the Vietnamese cultural knowledge and Vietnamese language subscales, and had the lowest scores on the American cultural identity, American cultural knowledge, and English language subscales. On average, personal network-recruited participants and those referred by them had the highest scores on the English language subscale and lived in zip codes with the lowest percentages of Vietnamese. Post-hoc analyses show that, compared to all other groups, participants recruited through CBOs had lived in the U.S. for the longest percentage of their lifetime (55.94) and had the lowest scores on the Vietnamese language subscale (3.76). Compared to those recruited through CBOs, those recruited through Facebook groups lived in zip codes with significantly lower percentages of Vietnamese (7.37 versus 3.24) and Asians (19.92 versus 12.70).

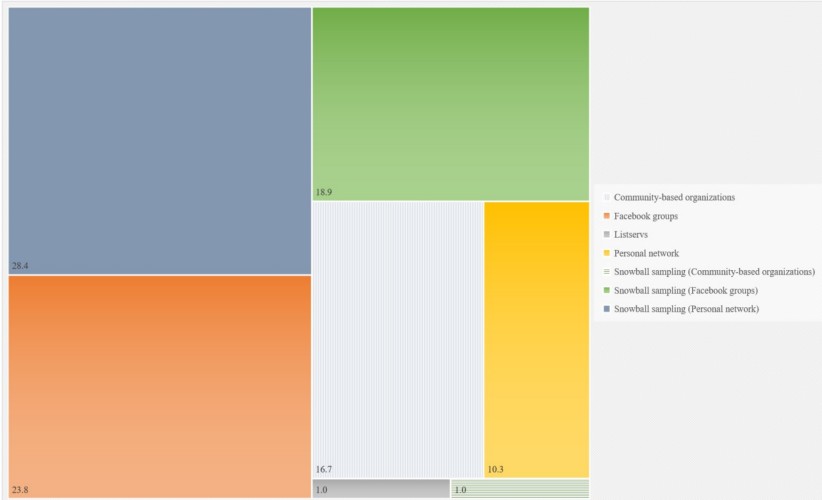

**Fig 3. Percentages of participants in the survey sample from different recruitment methods.**

**Table 2. Sociodemographic characteristics in relation to recruitment methods.**

| Sociodemographic characteristics | Total (n = 400)* | Community-based organizations (n = 68) | Facebook groups (n = 97) | Snowball sampling through Facebook (n = 77) | Personal network (n = 42) | Snowball sampling through personal network (n = 116) | p value |
|---|---|---|---|---|---|---|---|
| Age | 43.56 (5.74) | 45.00 (7.10) | 42.09 (4.91) | 44.06 (6.39) | 43.38 (5.51) | 43.66 (4.90) | .02 |
| Sex [a] (n = 397) | | | | | | | .001 |
| Male | 66 (16.6%) | 16 (23.9%) | 10 (10.5%) | 15 (19.5%) | 14 (33.3%) | 11 (9.5%) | |
| Female | 331 (83.4%) | 51 (76.1%) | 85 (89.5%) | 62 (80.5%) | 28 (66.7%) | 105 (90.5%) | |
| Parent's highest education level | | | | | | | < .001 |
| Less than a Bachelor's degree | 59 (14.8%) | 23 (33.8%) | 12 (12.4%) | 13 (16.9%) | 6 (14.3%) | 5 (4.3%) | |
| Bachelor's degree or higher | 341 (85.3%) | 45 (66.2%) | 85 (87.6%) | 64 (83.1%) | 36 (85.7%) | 111 (95.7%) | |
| Combined household income [b] (n = 349) | | | | | | | .04 |
| Less than $50,000 | 81 (23.2%) | 16 (24.6%) | 22 (24.4%) | 22 (32.4%) | 8 (21.6%) | 13 (14.6%) | |
| $50,000 to $100,000 | 122 (35.0%) | 19 (29.2%) | 38 (42.2%) | 26 (38.2%) | 12 (32.4%) | 27 (30.3%) | |
| $100,000 and above | 146 (41.8%) | 30 (46.2%) | 30 (33.3%) | 20 (29.4%) | 17 (46.0%) | 49 (55.1%) | |
| Child's country of birth | | | | | | | < .001 |
| Born in the U.S. | 192 (48.0%) | 53 (77.9%) | 29 (29.9%) | 28 (36.4%) | 24 (57.1%) | 58 (50.0%) | |
| Born outside of the U.S. | 208 (52.0%) | 15 (22.1%) | 68 (70.1%) | 49 (63.6%) | 18 (42.9%) | 58 (50.0%) | |

[a] Those who chose "Other" were coded as having missing data

[b] Those who chose "Prefer not to answer" were coded as having missing data

*Due to small cell sizes, the calculations in this table excluded eight participants who were recruited either through 1) listservs or 2) snowball sampling through CBOs.

**Mobile health usage.** We did not find any differences among recruitment methods regarding mobile health usage (**Table 3**). Overall, 77% reported daily use of a home computer to access the Internet and 92% reported daily use of a mobile device to access the Internet. Additionally, 89% and 88% of participants reported past 12-month use of a computer, smartphone, or other electronic means to look for health information for themselves and their children, respectively. A majority (60%) also reported past 12-month use of email or the Internet to communicate with doctors.

## Lessons learned in conducting online recruitment

**Handling fraudulent responses in online recruitment.** At the beginning of data collection, we created *public*, *non-unique links to the screener* that participants could fill out to determine their eligibility. Participants would be automatically redirected to the main survey if they met all the eligibility criteria. However, we encountered problematic responses after these links were publicly shared on Facebook. A closer examination of the initial responses and a review of the literature regarding fraudsters in online survey recruitment [58] led us to conclude that a majority of the initial responses were fraudulent.

These potentially fraudulent responses often followed these patterns: The provided email address generally indicated that both the participant's first and last name were not Vietnamese (e.g., Robert Anderson). Occasionally, the IP address was outside of the U.S. despite the survey response indicating that they lived in the U.S. Responses from the same participant about their sociodemographic characteristics (e.g., age, zip code) in the screener and in the main survey were considerably different. In addition, we detected illogical patterns of responses (e.g.,

**Table 3. Acculturation-related characteristics and mobile health usage in relation to recruitment methods.**

| Variables | Total (n = 400)[*] | Community-based organizations (n = 68) | Facebook groups (n = 97) | Snowball sampling through Facebook (n = 77) | Personal network (n = 42) | Snowball sampling through personal network (n = 116) | p value |
|---|---|---|---|---|---|---|---|
| **Acculturation-related characteristics** | | | | | | | |
| Percentage of life in the U.S. | 33.49 (25.37) | 55.94 (30.82) | 21.17 (16.59) | 25.39 (19.87) | 42.60 (27.01) | 32.70 (20.70) | < .001 |
| Ability to understand English medical information | | | | | | | .001 |
| Not at all to somewhat easy | 253 (63.3%) | 38 (55.9%) | 74 (76.3%) | 53 (68.8%) | 18 (42.9%) | 70 (60.3%) | |
| Very to extremely easy | 147 (36.8%) | 30 (44.1%) | 23 (23.7%) | 24 (31.2%) | 24 (57.1%) | 46 (39.7%) | |
| Asian American Multidimensional Acculturation Scale | | | | | | | |
| Vietnamese cultural identity [a] (n = 399) | 3.95 (0.74) | 3.79 (0.80) | 3.85 (0.83) | 4.11 (0.70) | 4.12 (0.59) | 3.97 (0.69) | .03 |
| Vietnamese cultural knowledge [a] (n = 399) | 3.90 (0.86) | 3.52 (1.18) | 3.98 (0.73) | 4.09 (0.73) | 3.86 (0.83) | 3.96 (0.77) | < .001 |
| Vietnamese language [a] (n = 399) | 4.38 (0.84) | 3.76 (1.36) | 4.62 (0.45) | 4.58 (0.52) | 4.28 (0.90) | 4.43 (0.64) | < .001 |
| Vietnamese food consumption [a] (n = 399) | 4.65 (0.55) | 4.62 (0.60) | 4.57 (0.61) | 4.73 (0.46) | 4.74 (0.37) | 4.64 (0.55) | .29 |
| American cultural identity | 2.89 (0.81) | 3.09 (0.91) | 2.77 (0.73) | 2.61 (0.85) | 3.24 (0.76) | 2.93 (0.71) | < .001 |
| American cultural knowledge | 2.60 (1.11) | 2.84 (1.22) | 2.39 (1.05) | 2.32 (1.09) | 3.07 (1.13) | 2.63 (1.01) | < .001 |
| English language | 3.42 (1.13) | 3.48 (1.34) | 3.11 (1.11) | 3.21 (1.12) | 3.96 (1.10) | 3.58 (0.91) | < .001 |
| American food consumption | 2.21 (1.20) | 2.62 (1.49) | 2.09 (1.11) | 1.94 (1.09) | 2.36 (1.22) | 2.21 (1.09) | .009 |
| Zip code-level percentage of Vietnamese [b] (n = 394) | 3.83 (8.42) | 7.37 (12.76) | 3.24 (8.05) | 5.46 (10.31) | 2.33 (4.15) | 1.71 (2.34) | < .001 |
| Zip code-level percentage of Asians [b] (n = 394) | 16.33 (14.53) | 19.92 (17.40) | 12.70 (13.33) | 19.18 (16.43) | 15.04 (12.80) | 15.87 (12.10) | .009 |
| **Mobile health usage** | | | | | | | |
| Daily use of mobile devices to access the Internet | | | | | | | |
| Use of computer at home [c] (n = 398) | 307 (77.1%) | 46 (67.7%) | 74 (76.3%) | 60 (77.9%) | 31 (73.8%) | 96 (84.2%) | .13 |
| Use of a mobile device [c] (n = 397) | 365 (91.9%) | 63 (92.7%) | 85 (87.6%) | 71 (93.4%) | 39 (92.9%) | 107 (93.9%) | .51 |
| Past 12-month use of mobile health | | | | | | | |
| Use of computer, smartphone, or electronic means to look for health or medical information for self [c] (n = 393) | 350 (89.1%) | 58 (86.6%) | 90 (92.8%) | 61 (81.3%) | 37 (88.1%) | 104 (92.9%) | .09 |
| Use of computer, smartphone, or electronic means to look for health or medical information for children [c] (n = 393) | 346 (88.0%) | 56 (83.6%) | 89 (91.8%) | 65 (86.7%) | 34 (81.0%) | 102 (91.1%) | .23 |
| Use of email or the Internet to communicate with a doctor or doctor's office [c] (n = 392) | 235 (60.0%) | 35 (52.2%) | 51 (54.3%) | 50 (66.7%) | 27 (64.3%) | 72 (63.2%) | .27 |

[*]Due to small cell sizes, the calculations in this table excluded eight participants who were recruited either through listservs or snowball sampling with CBOs

[a] One person who indicated their heritage culture was not Vietnamese culture was coded as having missing data

[b] Those whose zip code data are not available from the 2019 American Community Survey were coded as having missing data

[c] Those who chose "Prefer not to answer" were coded as having missing data

providing a year of immigration to the U.S. that was earlier than the year participant was born or much earlier than the end of the Vietnam War; responding that they had lived in Vietnam for the majority of their lives but indicated little fluency in Vietnamese).

Prior to launching the study, based on research team members' previous experiences conducting Internet-based surveys, we had configured the survey to only accept one response per IP address. All the potentially fraudulent responses had different, unique IP addresses. After detecting these responses, we then implemented the reCAPTCHA system embedded in SurveyGizmo/Alchemer and asked participants to check a box that said, "I am not a robot." We still, however, received problematic responses after implementing the reCAPTCHA system.

To ensure reliable data quality, we then decided to not continue with using public, non-unique links to the screener. Instead, we asked CBOs to disseminate information about our study and have eligible members contact us via emails. Similarly, with recruitment through Facebook groups, listservs, and snowball sampling, we asked participants to contact us directly. Study team members who were fluent in English and Vietnamese monitored the mailboxes. For participants who contacted us, we first verified where they learned of the study (e.g., certain CBOs, Facebook groups, or previous participants). We also reiterated the eligibility criteria. For those who confirmed where they learned of the study and that they met the criteria, we privately emailed them *unique, one-time use links to the screeners.*

**Perceived benefits and trust as facilitators of research participation.** Several participants remarked that they found our research project very meaningful. Specifically, many participants had limited knowledge of the HPV vaccine before completing the survey and reported that the project made them more aware of the benefits of the vaccine. For example, one interviewee stated: "I got my son vaccinated because I participated in your project and looked into the vaccine. Your research was a valuable source for me. Before that, I'd heard my friends talking about getting this vaccine to prevent cervical cancer, but it was all vague for me. . . So, your project came just in time." Another echoed the same sentiment: "Before filling out your survey, I didn't know that boys should get the HPV vaccine too." In addition, a participant said: "This project is useful for me as it has helped me pay more attention to cervical cancer. I have heard about cervical cancer, but I have not paid much attention to the vaccine".

Snowball sampling appears to be a successful recruitment method. Many participants who found the project meaningful were enthusiastic in referring their eligible acquaintances, and several of those who were referred mentioned that they felt motivated to participate because their acquaintances had completed the survey. The information about the project that was included in introductory emails and messages also helped generate trust. The first author had included her name, credentials, past experiences with research in Vietnamese health, study funders, IRB approval, and contact information, in addition to study information (e.g., purpose, eligibility criteria, procedure, and compensation). Some participants noted that having such comprehensive information helped them trust that the responses they were providing would be used for research and health programs for Vietnamese communities and not for other purposes (e.g., for-profit purposes).

## Discussion

In this paper, we described the results of different online recruitment strategies of U.S. Vietnamese for a health research project as well as lessons drawn from conducting this project. To our knowledge, this paper is the first to present a variety of online recruitment strategies for U.S. Vietnamese and compare characteristics of participants among methods. While the overall response rate was high (72%), response rates varied among different recruitment methods. In addition, we found significant between-group differences in all sociodemographic characteristics and almost all acculturation-related characteristics. Mobile health use, however, appeared similar among recruitment methods. A considerable challenge during the recruitment process

was encountering fraudulent responses. Moreover, participants' perceived benefits and trust encouraged their research participation.

In general, response rates from CBOs were the lowest (e.g., 52% for Vietnamese-serving CBOs). While we made efforts to contact 320 CBOs (including VSAs) for recruitment, we received completed responses from participants from only 28 of those (i.e. less than 10%). A previous study noted similar difficulties in online recruitment of Korean-Americans through communities and groups catering to this population [32]. Of 422 Korean-American communities and groups contacted, only 72 agreed to announce the study through their websites and email lists, and only 13 participants were recruited into the study [32]. Research using online recruitment of Asian populations, especially those planning for a moderate sample size, should contact a high number of CBOs given these low observed response rates. Establishing relationships with gatekeepers (e.g., webmasters of online groups, directors of CBOs) may also be essential to facilitating recruitment [31]. Two studies conducted during the COVID-19 pandemic (one with residents in rural areas in Mozambique [59] and one with mothers who had experienced domestic violence in Canada [60]) have also found that engagement with CBOs and stakeholders continued to be crucial for participant recruitment, even when researchers moved to remote data collection. At the same time, the pandemic likely created great disruptions to the activities of CBOs and exacerbated healthcare and social needs from community members [61], all of which would put a strain on the operations of CBOs. This situation may explain the low response rates from CBOs in our study and researchers should be mindful of the limited time and effort that CBOs may be able to offer for research recruitment during a time of crisis.

In contrast to recruitment with CBOs, we received completed responses from participants from all Facebook groups where we advertised the study, with a high overall response rate (87%). Moreover, response rates for snowball sampling recruitment from either Facebook or the first author's personal network were also high (76% and 73%, respectively). These results suggest the high feasibility of recruitment through Facebook groups or snowball sampling for U.S. Vietnamese populations during the COVID-19 pandemic.

Participants recruited through CBOs had lived in the U.S. for the longest duration and had the lowest Vietnamese language ability. When reviewing websites of Vietnamese-serving CBOs during recruitment, we noted that many were established between 1976 and mid-1990s. This period followed the Vietnam War, when hundreds of thousands of Vietnamese refugees left for the U.S.; it was also prior to the bilateral normalization of U.S.–Vietnam relations [62, 63]. Participants recruited through CBOs may reflect the demographic of the groups that emigrated out of Vietnam in the earlier waves. Compared to those recruited through Facebook groups, those recruited through CBOs also lived in areas with higher concentration of Vietnamese and Asians, which are likely also places where CBOs are based. Moreover, given that Facebook groups that we selected operated in Vietnamese and were geared towards advice for Vietnamese living in the U.S., it is understandable that participants recruited through Facebook groups had lived in the U.S. for shorter durations, had higher Vietnamese language ability and lower English language ability, and had lower scores on American acculturation subscales.

Researchers looking to replicate these recruitment methods for future research studies in Vietnamese populations should be mindful of subgroup variability in terms of sociodemographic characteristics and acculturation-related characteristics, even within a cultural population (e.g., U.S. Vietnamese). For health promotion efforts in particular, these differences point to the need to consider tailored approaches to cultural appropriateness [64]. Knowing a person's educational background, level of Vietnamese cultural identity, degree of Vietnamese cultural knowledge, and ability to speak Vietnamese or English will facilitate crafting effective and

resonating health messages [64]. For example, in our sample, the majority of Facebook groups-recruited participants and those referred by them likely would require messages and programs in Vietnamese that leverage Vietnamese cultural information. In contrast, a considerable number of those recruited through CBOs and the first author's personal network may prefer information in Vietnamese that incorporates Vietnamese culture but may also be open to receiving information in English given their higher English proficiency. Moreover, findings from our study highlight the complexity of the relationships between participants' Vietnamese and American acculturation degrees, and consequently, the importance of including multidimensional acculturation-related measurements [65, 66]. For example, lower Vietnamese acculturation does not necessarily mean higher American acculturation (and vice-versa), as on average, those recruited through the first author's personal network had the highest scores for both Vietnamese and American cultural identity subscales.

To our knowledge, no research has examined mobile health usage among U.S. Vietnamese. Our study found no differences in mobile health usage among recruitment methods. Overall, more participants reported daily use of a mobile device than daily use of a computer to access the Internet, which has implications for the delivery mediums of research and interventions. In addition, nearly 9 out of 10 of participants reported past 12-month use of a computer, smartphone, or electronic means to look for health information for themselves and their children. This finding points to the potential for implementing mobile-based interventions to address health issues in U.S. Vietnamese.

We encountered fraudulent responses in the beginning, which we resolved by switching to manually verifying participants via emails first and then sending them unique screener links. While we believe this process helped ensure reliable data quality, it was also time-consuming and may have discouraged some participants (e.g., due to the need to send emails before they could participate in the study). Previous research with non-Asian populations has raised issues with fraudulent responses in online recruitment [67–69]. A growing body of literature is focusing on strategies to implement quality checks and identify fraudulent or automated responses in online survey research [58, 70–74]. These threats may be mitigated through methods such as assessing response coherence and consistency, including open-ended questions, and adding attention and logic checks [58, 70–74]. Aside from questionnaire data, researchers could also examine other information provided (e.g., emails and phone numbers) and look for similar/duplicate items or validate participants via phone prior to accepting the data [58]. Having cultural knowledge of the targeted population was useful for our fraud detection, as we noticed patterns of non-Vietnamese names in email addresses and illogical responses.

Previous studies with Asian communities in the U.S. (including both online and non-online recruitment) have highlighted the value of cultural concordance in facilitating recruitment. For example, Im and colleagues described how they were able to contact Chinese-American communities and groups only after working with a Chinese-American research assistant, but not research assistants of other Asian subgroups [31]. Mukherjea and colleagues also noted that respondents in their study had preferences for interviewers from the same South Asian regions, and that effective "cultural research brokers" are those that not only visibly and linguistically represented the community but also possessed credentials (e.g., advanced degrees) held in high regard by community members [21]. Other studies have also reported a preference for culturally matched research personnel in Hawaiian and Filipino populations [75, 76]. In our study, aside from being U.S. Vietnamese and fluent in the language, the first author had shared her credentials and past experiences with research in Vietnamese health with participants, which increased some participants' trust in the project. In addition, a few also perceived gaining more knowledge about the HPV vaccine as an indirect benefit from study participation, which made them enthusiastic about referring others to the study. Recent systematic

reviews have also documented knowledge gain and wanting to help others as facilitators of participation in research [77, 78].

While, overall, our project had relative success in recruiting participants remotely during the COVID-19 pandemic, we note that remote data collection that relies on digital technologies is not always appropriate in every context and situation [79–81]. Researchers should consider how the pandemic may have affected the participants' well-being and their ability to participate in research [81]. In addition, while our participants did not voice concerns about technological barriers, multiple studies have demonstrated inequities in access to and ability to use mobile devices (e.g., smartphones, tablets, or laptops) for research participation along with challenges related to poor connectivity. Specifically, such issues have been documented in studies conducted during the COVID-19 pandemic with older adults in the U.S. and Italy [82, 83], youth volunteers in Sudan [84], and adults in Malawi and the United Kingdom [81, 84].

## Limitations

Our results should be interpreted in light of a few limitations. We did not collect data on sociodemographic characteristics, acculturation, or mobile health usage of those who refused participation; therefore, we could not detect whether there were significant differences between those who chose to participate and those who did not. Additionally, we did not collect data on reasons why individuals refused participation.

While recruiting participants from existing social networks of research personnel has been used in several past studies with minority populations [69, 85, 86], including Vietnamese [87], given the prevalence of homophily [88] in social networks (e.g., "tendency for friendships to form between those who are alike in some designated respect") [89], it is possible that personal network-recruited participants would share several similarities to the first author. For example, homophily may explain why personal network-recruited participants reported on average the highest level of English language proficiency. Future studies using similar methods should take into account the characteristics of the recruiters when determining the generalizability of research findings. Likewise, snowball sampling can lead to an overrepresentation of participants who share similarities or have larger social networks, consequently creating a sample that is unbalanced in several characteristics [50, 86, 90]. Future research can consider employing respondent-driven sampling, which uses the chain-referral sample to make estimates about the social networks, and then uses the information about the social networks to derive population proportions [90, 91]. Moreover, due to limited resources, for recruitment using Facebook, we posted about the study in relevant Facebook groups instead of using targeted Facebook advertising. While our approach had the advantage of bearing no direct costs, only members of the chosen Facebook groups had access to information about the study. The use of targeted Facebook advertising could have increased our reach to users who were not members of these groups.

Finally, the lack of sampling frames or population estimates for U.S. Vietnamese parents of adolescents prevents us from comparing estimates of our sample and those from the population. Our sample of parents of adolescents, on average, has higher socioeconomic status (e.g., income and education) compared to the 2019 American Community Survey data for all Vietnamese in the U.S. [52].

## Conclusions

This paper presents a case study of multi-modal internet-based recruitment of U.S. Vietnamese into research. We found differences in sociodemographic and acculturation-related characteristics of participants drawn through different sources. Participants reported high mobile

health usage among recruitment methods. Recruitment through Facebook groups and snowball sampling yielded high response rates, suggesting that these are feasible strategies to recruit U.S. Vietnamese. Perceived benefits and trust appear to be facilitators to study recruitment. Special attention should be paid to recruitment with CBOs and handling fraudulent responses. Our findings can inform future multi-modal data collection efforts for research with U.S. Vietnamese, Asian communities in the U.S., and other understudied populations in an international context.

## Author Contributions

**Conceptualization:** Milkie Vu.

**Data curation:** Milkie Vu, Robert A. Bednarczyk, Cam Escoffery, Tien T. Nguyen, Carla J. Berg.

**Formal analysis:** Milkie Vu.

**Funding acquisition:** Milkie Vu.

**Investigation:** Milkie Vu, Robert A. Bednarczyk, Cam Escoffery, Danny Ta, Tien T. Nguyen.

**Methodology:** Milkie Vu, Victoria N. Huynh, Tien T. Nguyen, Carla J. Berg.

**Project administration:** Milkie Vu, Victoria N. Huynh.

**Resources:** Milkie Vu, Victoria N. Huynh.

**Software:** Milkie Vu.

**Supervision:** Milkie Vu.

**Validation:** Milkie Vu, Danny Ta, Tien T. Nguyen.

**Visualization:** Milkie Vu.

**Writing – original draft:** Milkie Vu.

**Writing – review & editing:** Milkie Vu, Victoria N. Huynh, Robert A. Bednarczyk, Cam Escoffery, Danny Ta, Tien T. Nguyen, Carla J. Berg.

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
