## [Decision Letter · Decision Letter 0]

14 Jun 2021

PONE-D-21-14089

Experience and lessons learned from multi-modal internet-based recruitment of U.S. Vietnamese into research

PLOS ONE

Dear Authors,

Thank you for submitting your manuscript to PLOS ONE. After careful consideration, we feel that it has merit but does not fully meet PLOS ONE’s publication criteria as it currently stands. Therefore, we invite you to submit a revised version of the manuscript that addresses the points raised during the review process.

We look forward to receiving your revised manuscript.

Kind regards,

Marcel Pikhart

Academic Editor

PLOS ONE

Journal Requirements:

Reviewers' comments:

Reviewer's Responses to Questions

**Comments to the Author**

1. Is the manuscript technically sound, and do the data support the conclusions?

Reviewer #1: Yes

Reviewer #2: Yes

Reviewer #3: Yes

2. Has the statistical analysis been performed appropriately and rigorously? 

Reviewer #1: Yes

Reviewer #2: Yes

Reviewer #3: No

3. Have the authors made all data underlying the findings in their manuscript fully available?

Reviewer #1: Yes

Reviewer #2: No

Reviewer #3: No

4. Is the manuscript presented in an intelligible fashion and written in standard English?

Reviewer #1: Yes

Reviewer #2: Yes

Reviewer #3: Yes

5. Review Comments to the Author

Reviewer #1: Thank you for giving me this opportunity to review the paper. I completely agree with the authors that studying the Asian American population is critical and underdeveloped in current research. This paper could potentially be valuable in achieving this goal, but I do think it needs clarification on the following points.

1. The paper’s focus switches between the Asian population and the Vietnamese-Asian population. My largest question is which of these findings on recruitment are specific to studying the Vietnamese-Asian population, which of these are specific to studying the Asian population, and which of the findings are applicable to studying all people? For example, the paper mentions this paragraph on the lessons regarding snowball sampling: “Snowball sampling appears to be a successful recruitment method. Many participants who found the project meaningful were enthusiastic in referring their eligible acquaintances, and several of those who were referred mentioned that they felt motivated to participate because their acquaintances had completed the survey” (p. 21). Is this lesson specific to Asian recruitment? Or is it applicable to non-Asian Americans as well? Put differently, if we used snowball sampling to recruit Latino/a people, would we also expect participants to find the project meaningful and share their enthusiasm? I think the paper would be stronger if the authors cut the parts that are not particularly relevant for the recruitment of the Asian population. After all, the study of Asian Americans is the biggest value of this paper.

2. I’m not sure about why mobile health was included as one of the research dimensions, as the other aspects (i.e., demographics and acculturation) are more relevant to this paper’s focus on ethnicity. Are there special things that researchers should be aware of when concerning Asian people’s mobile health? If so, what are they and how are these issues related to recruitment? I think the reader would appreciate some more clarifications.

3. My third question is the validity of the findings regarding the “personal networks” recruitment method, which is substantial in the paper. Specifically, I’m wondering whether the findings are because of the characteristics of the recruiter. Recruitment through personal networks depends highly on who is the recruiter. The people recruited would likely share similarities with the recruiter (in other words, “homophily” in the network literature). In this case, the recruiters are highly educated researchers, and thus one would expect a network of highly educated people. It is thus no surprise that this subsample had the highest scores on the English language subscale (p. 19). However, is this finding generalizable to other researchers who use the method? The other findings are similarly suspect. For instance, if a researcher asked a community organization with low education members to recruit participants, would one expect the same results (either regarding English, zip codes, or other patterns)?

If the authors can successfully clean up some of the confusions, the paper would make a contribution to the literature on studying the Asian American population.

Reviewer #2: Abstract

The ABSTRACT is well written, addressing the investigated problems intelligently.

Introduction

The INTRODUCTION is also well-structured, highlighting the problems of drawing samples from online data collection approach eloquently. Besides, the existing literature is well-documented to justify the research questions raised by the authors.

Materials and methods

The reviewer really appreciates the METHODOLOGY, as the authors have already addressed the limitations of doing this research. However, the authors are requested to delete “: Survey” from the Recruitment procedures as the study is based on survey only. Furthermore, the authors are requested to specify the sampling procedure for ‘community-based organizations,’ ‘Facebook groups’ and so on, except for ‘snowball sampling’ group. The authors are advised to use a sub-heading ‘Measurement’ under which they are requested to place ‘Sociodemographic characteristics,’ ‘Acculturation-related characteristics’ and ‘Mobile health usage,’ instead of repeating Measurement for each sub-heading.

It is not clear how did the authors select the participants for interviews, how and what tools they used to interview for participants for qualitative interviews.

The authors are requested to specify the procedures of interviewing the participants for qualitative interview – whether it was face-to-face, or telephonic or other procedure of data collection maintain social distancing.

Results

The results are well presented and pointed out some critical issues regarding the short-falls and advantages of using internet-based recruitment of participants during the COVID-19.

Discussion

The reviewer also appreciates the DISCUSSION penned by the authors. Yet, the DISCUSSION lacks recent empirical evidence regarding the problems that are critical to understand the ongoing situation of recruiting participants during the COVID-19 pandemic.

Conclusion

Th author are requested to point out whether this study is generalizable to other countries or not, and how other nations could cope with the emerging problems of the online recruitment during the COVID-19 pandemic.

Others

The authors are requested to follow the PLoS ONE reference style properly. Many citations did not comply with the reference style suggested by PLoS ONE.

Reviewer #3: The goal of this paper is to assess different forms of recruiting Vietnamese participants online. The paper does a good job in describing the many different types of recruiting the researchers did. However, I see two main ways that the paper can be improved: (1) The authors could do more to analyze the strengths and drawbacks of each method in relation to one another. For example, they have a section that says that snowball sampling is a good method to use, however, there is no discussion of the drawbacks of this method. Importantly, there is no discussion in the paper of what particular sub-sample of the Vietnamese population the authors are able to recruit, and the drawbacks of not being able to know how representative any of these subsamples are of the overall Vietnamese population. If the authors believe that getting representative samples is simply too difficult, it would be useful to understand more about why – is it because the population is very small, is it because this is a group that is just very hard to reach, etc. Or, perhaps these findings shed light on how representative samples of Vietnamese groups in the US can be reached. (2) The paper seems to be about health and health information, but there is very little description or tie-in to health. There are some brief mentions of the HPV study, however, the paper would be strengthened by a richer literature review of how better health information or services could benefit this population, and what is learned in particular about these various types of recruiting for challenges in health among Vietnamese populations.

A few more specific comments are below:

1. Intro:

a. The authors specify that “Asian-Americans” refers to Vietnamese, Korean, Chinese individuals – it is not clear if that is in the context of this paper alone, or across studies. This term is often used for other groups also. The statistics in the first paragraph do not make clear which subgroups are included and which are not. Perhaps change to “(e.g. Vietnamese, Korean, Chinese, etc.).

b. The motivation for using community surveys to reach Asian language speaking individuals is noted, but what fraction of Asians is this?

c. The motivation that it is difficult to reach this group, that online data collection is going to be hard because community events are not a possibility anymore due to COVID – this is all important. It would be helpful to have some background on studies that show information on how this group of non-English speaking Asians is different – like health utilization? Discrimination? Income? Social service usage? Just saying that Asians are the fastest growing population is a bit odd if then the focus is on this non-English speaking group.

d. And then the focus become Vietnamese. This makes sense given that they have lower English proficiency, lower income and education, etc. But the authors link this to barriers to health service utilization. The connection to health services is not made clear. That should be fleshed out much more, that the primary concern here is health service utilization.

e. Very little time in the intro is spent describing the HPV vaccine study. It would be good to flesh this out more too, to talk about how not having information from non-English speaking individuals could lead to incorrect results in a study like this.

2. Methods:

a. If only one parent per household was allowed to participate, would they have had the more English fluent person participate? Does this skew results?

3. Results:

a. It would be helpful to know how those who refused looked in terms of SES and other characteristics. Why did they refuse?

b. The authors describe the differences in SES and in acculturation-related characteristics, but do not explore what this might mean to the results. A fuller analysis would strengthen the paper.

c. In Table 3 - There is no description of the scale used for the acculturation measure, thus it is unclear what a score of 3.95 or 4.65 mean in relation to one another.

d. In Table 3 - Again here it would be helpful to know exactly how “Asian” is defined

e. In Table 3 and in text – consider separating out “mobile health usage” and “mobile phone usage”

f. It is not clear where the results of the chi-square tests/tukey tests are reported in looking at differences across sociodemographic and other characteristics, or differences in pairs of means.

g. Also, it would be great to have more details from the qualitative study. I believe there is only a single quote in the paper, but more details from the qualitative work could make the paper

6. PLOS authors have the option to publish the peer review history of their article (what does this mean?). If published, this will include your full peer review and any attached files.

Reviewer #1: No

Reviewer #2: **Yes: **Md. Tanvir Hossain

Reviewer #3: No

---

## [Author Response · Author response to Decision Letter 0]

17 Jul 2021

AUTHORS: We have revised accordingly to ensure that the manuscript meets the style requirements. 

AUTHORS: We have revised accordingly to make sure the correct grant numbers are reflected. 

b) If there are no restrictions, please upload the minimal anonymized data set necessary to replicate your study findings as either Supporting Information files or to a stable, public repository and provide us with the relevant URLs, DOIs, or accession numbers. For a list of acceptable repositories, please see http://journals.plos.org/plosone/s/data-availability#loc-recommended-repositories. We will update your Data Availability statement on your behalf to reflect the information you provide.

AUTHORS: We have clarified: "The data that support the findings of this study are available upon reasonable request and with the approval of Emory University Institutional Review Board (IRB 00111688). The data are not publicly available as it contains information that could compromise the privacy of research participants. In particular, given that participants are Vietnamese parents of adolescents (a small, specific population), and that the dataset contains their zip code, making the data available can risk the possibility of participants being identified. Please see more information here: http://www.irb.emory.edu/documents/phi_identifiers.pdf

Please contact the lead author, MV, as well as the Institutional Review Board at Emory University, with any request for data access (milkie.vu@emory.edu and IRB@emory.edu)." 

Comments to the Author

1. Is the manuscript technically sound, and do the data support the conclusions?

Reviewer #1: Yes

Reviewer #2: Yes

Reviewer #3: Yes

2. Has the statistical analysis been performed appropriately and rigorously?

Reviewer #1: Yes

Reviewer #2: Yes

Reviewer #3: No

3. Have the authors made all data underlying the findings in their manuscript fully available?

Reviewer #1: Yes

Reviewer #2: No

Reviewer #3: No

4. Is the manuscript presented in an intelligible fashion and written in standard English?

Reviewer #1: Yes

Reviewer #2: Yes

Reviewer #3: Yes

5. Review Comments to the Author

Reviewer #1: Thank you for giving me this opportunity to review the paper. I completely agree with the authors that studying the Asian American population is critical and underdeveloped in current research. This paper could potentially be valuable in achieving this goal, but I do think it needs clarification on the following points.

1. The paper’s focus switches between the Asian population and the Vietnamese-Asian population. My largest question is which of these findings on recruitment are specific to studying the Vietnamese-Asian population, which of these are specific to studying the Asian population, and which of the findings are applicable to studying all people? For example, the paper mentions this paragraph on the lessons regarding snowball sampling: “Snowball sampling appears to be a successful recruitment method. Many participants who found the project meaningful were enthusiastic in referring their eligible acquaintances, and several of those who were referred mentioned that they felt motivated to participate because their acquaintances had completed the survey” (p. 21). Is this lesson specific to Asian recruitment? Or is it applicable to non-Asian Americans as well? Put differently, if we used snowball sampling to recruit Latino/a people, would we also expect participants to find the project meaningful and share their enthusiasm? I think the paper would be stronger if the authors cut the parts that are not particularly relevant for the recruitment of the Asian population. After all, the study of Asian Americans is the biggest value of this paper.

AUTHORS: Thank you very much for these comments. To address this point, we have removed two paragraphs from the Introduction and Discussion that are broad and not specific to our populations. At the same time, we want to emphasize that there is very little research that describes online recruitment strategies specifically for U.S. Vietnamese. Consequently, we needed to draw on the body of research with other Asian subgroups and consult other literature on online recruitment with general populations. Our manuscript, as you suggested, provides information on findings applicable to the recruitment of U.S. Vietnamese and attempts to contextualize these findings in a broader literature on the recruitment of Asian-Americans into research studies.

Several findings, such as the sociodemographic and acculturation-related differences between those recruited via community-based organizations and those recruited via Facebook groups, are more specific to the history of migration and social patterns of U.S. Vietnamese. Findings related to low response rates from community-based organizations and the role of cultural concordance in facilitating recruitment have been noted in other studies with Asian-Americans. Issues with encountering fraudulent responses are a growing challenge with online recruitment and have been reported in research with different populations. As you suggested, research with other populations has also documented the effectiveness of snowball sampling. 

We believe that in this manuscript, it is important to report on findings specific to the recruitment of U.S. Vietnamese as well as several findings that may be generalized to the recruitment of Asians and even other populations. For example, it is critical for researchers embarking on future studies of U.S. Vietnamese or Asian-Americans to know about possible fraudulent responses in online research so they can take appropriate measures to counter this threat. In the manuscript, we reported on the relatively higher response rates from snowball sampling recruitment compared to recruitment from community organizations and commented on the utility of snowball sampling. Again, this information will be useful for others who are conducting future research with U.S. Vietnamese or Asian-Americans. 

2. I’m not sure about why mobile health was included as one of the research dimensions, as the other aspects (i.e., demographics and acculturation) are more relevant to this paper’s focus on ethnicity. Are there special things that researchers should be aware of when concerning Asian people’s mobile health? If so, what are they and how are these issues related to recruitment? I think the reader would appreciate some more clarifications.

AUTHORS: Thank you for these questions. We have added more information about why we collected data on mobile health usage: "Given the critical role of digital technologies in facilitating data collection during the COVID-19 pandemic, these data can indicate whether access to the Internet (and, consequently, ability to participate in online data collection) differs across participants from different recruitment methods and potentially explain differences in response rates. In addition, they can also provide information on the possible receptiveness of participants to research and interventions leveraging mobile health technologies." 

3. My third question is the validity of the findings regarding the “personal networks” recruitment method, which is substantial in the paper. Specifically, I’m wondering whether the findings are because of the characteristics of the recruiter. Recruitment through personal networks depends highly on who is the recruiter. The people recruited would likely share similarities with the recruiter (in other words, “homophily” in the network literature). In this case, the recruiters are highly educated researchers, and thus one would expect a network of highly educated people. It is thus no surprise that this subsample had the highest scores on the English language subscale (p. 19). However, is this finding generalizable to other researchers who use the method? The other findings are similarly suspect. For instance, if a researcher asked a community organization with low education members to recruit participants, would one expect the same results (either regarding English, zip codes, or other patterns)?

AUTHORS: Thank you for these suggestions. Out of our sample of 408 participants, around 10% (n=42) were recruited from the first author's personal network. We agree with your point about homophily in social network research and have added to the Limitation section: "While recruiting participants from existing social networks of research personnel has been used in several past studies with minority populations (1–3), including Vietnamese (4), given the prevalence of homophily (5) in social networks (e.g., “tendency for friendships to form between those who are alike in some designated respect”) (6), it is possible that personal network-recruited participants would share several similarities to the first author. For example, homophily may explain why personal network-recruited participants reported on average the highest level of English language proficiency. Future studies using similar methods should take into account the characteristics of the recruiters when determining the generalizability of research findings." 

In our original submission, we also noted several characteristics of the first author that may have positively influenced the recruitment process, such as her experience in research with U.S. Vietnamese and her culturally matched background with the research population. 

Regarding your point about recruiting through community organizations, this was one of the approaches we had used. To achieve a diversity of participants, we had attempted to contact an extensive number of organizations (115 Vietnamese-serving CBOs, 91 Asian-serving CBOs, and 114 VSAs). These organizations then helped connect us to eligible and interested study participants. We expect that if another researcher replicates such methodology for a study with U.S. Vietnamese, they would find similar results regarding the sociodemographic and acculturation of CBO-recruited participants. 

If the authors can successfully clean up some of the confusions, the paper would make a contribution to the literature on studying the Asian American population.

Reviewer #2: 

Abstract

The ABSTRACT is well written, addressing the investigated problems intelligently.

Introduction

The INTRODUCTION is also well-structured, highlighting the problems of drawing samples from online data collection approach eloquently. Besides, the existing literature is well-documented to justify the research questions raised by the authors.

Materials and methods

The reviewer really appreciates the METHODOLOGY, as the authors have already addressed the limitations of doing this research. However, the authors are requested to delete “: Survey” from the Recruitment procedures as the study is based on survey only. Furthermore, the authors are requested to specify the sampling procedure for ‘community-based organizations,’ ‘Facebook groups’ and so on, except for ‘snowball sampling’ group. The authors are advised to use a sub-heading ‘Measurement’ under which they are requested to place ‘Sociodemographic characteristics,’ ‘Acculturation-related characteristics’ and ‘Mobile health usage,’ instead of repeating Measurement for each sub-heading.

AUTHORS: Thank you for these suggestions. We have added: "Convenience sampling was used for all recruitment methods." 

We specified two subheadings "Recruitment procedures: Survey" and "Recruitment procedures: Interviews" to distinguish between recruitment for the survey and recruitment of a subset of survey respondents for the interviews. Per your comment about the subheadings for "Measurements," we have revised accordingly and also attended to PlosOne's manuscript convention in which there are only three levels of subheadings.

It is not clear how did the authors select the participants for interviews, how and what tools they used to interview for participants for qualitative interviews. The authors are requested to specify the procedures of interviewing the participants for qualitative interview – whether it was face-to-face, or telephonic or other procedure of data collection maintain social distancing.

AUTHORS: We have added: "Between November 2020 and February 2021, we invited a subset of participants who had already completed the survey to participate in in-depth, semi-structured qualitative interviews. We purposively sampled participants and stratified the sample of interviewees by their adolescent child's sex (female versus male) and HPV vaccination status. For selected potential interviewees, we send each of them an email with information about the interviews and a consent form. Depending on participants' preferences, interviews were conducted in either Vietnamese or English and via telephone or the Zoom platform."

Results

The results are well presented and pointed out some critical issues regarding the short-falls and advantages of using internet-based recruitment of participants during the COVID-19.

Discussion

The reviewer also appreciates the DISCUSSION penned by the authors. Yet, the DISCUSSION lacks recent empirical evidence regarding the problems that are critical to understand the ongoing situation of recruiting participants during the COVID-19 pandemic.

Conclusion

The author are requested to point out whether this study is generalizable to other countries or not, and how other nations could cope with the emerging problems of the online recruitment during the COVID-19 pandemic.

AUTHORS: To address these comments, we have added two paragraphs to the Discussion: "Two studies conducted during the COVID-19 pandemic (one with residents in rural areas in Mozambique (7) and one with mothers who had experienced domestic violence in Canada (8)) have also found that engagement with CBOs and stakeholders continued to be crucial for participant recruitment, even when researchers moved to remote data collection. At the same time, the pandemic likely created great disruptions to the activities of CBOs and exacerbated healthcare and social needs from community members (9), all of which would put a strain on the operations of CBOs. This situation may explain the low response rates from CBOs in our study. Future research should be mindful of the limited time and efforts that CBOs may be able to offer for research recruitment during a time of crisis."

"While, overall, our project had relative success in recruiting participants remotely during the COVID-19 pandemic, we note that remote data collection that relies on digital technologies is not always appropriate in every context and situation (10–12). Researchers should consider how the pandemic may have affected the participants' well-being and their ability to participate in research (12). In addition, while our participants did not voice concerns about technological barriers, multiple studies had demonstrated inequities in access to and ability to use mobile devices (e.g., smartphones, tablets, or laptops) for research participation along with challenges related to poor connectivity. Specifically, such issues have been documented in studies conducted during the COVID-19 pandemic with older adults in the U.S. and Italy (13,14), youth volunteers in Sudan (15), and adults in Malawi and the United Kingdom (12,15)."

 We have also added to the Conclusion: "Our findings can inform future multi-modal data collection efforts for research with U.S. Vietnamese, Asian communities in the U.S., and other understudied populations in an international context."

Others

The authors are requested to follow the PLoS ONE reference style properly. Many citations did not comply with the reference style suggested by PLoS ONE.

AUTHORS: We used a reference software (Mendeley) to manage our references. We have reviewed and made sure the paper adheres to the PLoSOne reference style. 

Reviewer #3: The goal of this paper is to assess different forms of recruiting Vietnamese participants online. The paper does a good job in describing the many different types of recruiting the researchers did. However, I see two main ways that the paper can be improved: 

(1) The authors could do more to analyze the strengths and drawbacks of each method in relation to one another. For example, they have a section that says that snowball sampling is a good method to use, however, there is no discussion of the drawbacks of this method. 

AUTHORS: Thank you for these remarks. We have added: "While recruiting participants from existing social networks of research personnel has been used in several past studies with minority populations (1–3), including Vietnamese (4), given the prevalence of homophily (5) in social networks (e.g., “tendency for friendships to form between those who are alike in some designated respect”) (6), it is possible that personal network-recruited participants would share several similarities to the first author. For example, homophily may explain why personal network-recruited participants reported on average the highest level of English language proficiency. Future studies using similar methods should take into account the characteristics of the recruiters when determining the generalizability of research findings. Likewise, snowball sampling can lead to an overrepresentation of participants who share similarities or have larger social networks, consequently creating a sample that is unbalanced in several characteristics (3,16,17). Future research can consider employing respondent-driven sampling, which uses the chain-referral sample to make estimates about the social networks, and then uses the information about the social networks to derive population proportions (17,18). Moreover, due to limited resources, for recruitment using Facebook, we posted about the study in relevant Facebook groups instead of using targeted Facebook advertising. While our approach had the advantage of bearing no direct costs, only members of the chosen Facebook groups had access to information about the study. The use of targeted Facebook advertising could have increased our reach to users who were not members of these groups."

 In the original submission, we had also discussed some of the advantages and drawbacks of recruiting from the first author's personal network and from community-based organizations. 

Importantly, there is no discussion in the paper of what particular sub-sample of the Vietnamese population the authors are able to recruit, and the drawbacks of not being able to know how representative any of these subsamples are of the overall Vietnamese population. If the authors believe that getting representative samples is simply too difficult, it would be useful to understand more about why – is it because the population is very small, is it because this is a group that is just very hard to reach, etc. Or, perhaps these findings shed light on how representative samples of Vietnamese groups in the US can be reached. 

AUTHORS: Thank you for these questions. We are addressing them in our reply to your point in 1.e. (please see below). In addition, we have also added: "The lack of sampling frames or population estimates for U.S. Vietnamese parents of adolescents prevents us from comparing estimates of our sample and those from the population. Our sample of parents of adolescents, on average, has higher socioeconomic status (e.g., income and education) compared to the 2019 American Community Survey data for all Vietnamese in the U.S. (19)"

(2) The paper seems to be about health and health information, but there is very little description or tie-in to health. There are some brief mentions of the HPV study, however, the paper would be strengthened by a richer literature review of how better health information or services could benefit this population, and what is learned in particular about these various types of recruiting for challenges in health among Vietnamese populations.

AUTHORS: Thank you for these questions. We are addressing them in our reply to your point in 1.e. (please see below). 

A few more specific comments are below:

1. Intro:

a. The authors specify that “Asian-Americans” refers to Vietnamese, Korean, Chinese individuals – it is not clear if that is in the context of this paper alone, or across studies. This term is often used for other groups also. The statistics in the first paragraph do not make clear which subgroups are included and which are not. Perhaps change to “(e.g. Vietnamese, Korean, Chinese, etc.)

AUTHORS: Thank you for this comment. To avoid misunderstanding, we have changed this sentence to: "Moreover, when Asian-American participants are included, research typically aggregates Asian subgroups into one single category (20–22) instead of providing data for separate subgroups."

b. The motivation for using community surveys to reach Asian language speaking individuals is noted, but what fraction of Asians is this?

AUTHORS: We have added: "According to the 2019 American Community Survey, 74% and 31% of the Asian population in the U.S. reported that they spoke a language other than English and that they spoke English less than "very well," respectively (23)."

c. The motivation that it is difficult to reach this group, that online data collection is going to be hard because community events are not a possibility anymore due to COVID – this is all important. It would be helpful to have some background on studies that show information on how this group of non-English speaking Asians is different – like health utilization? Discrimination? Income? Social service usage? Just saying that Asians are the fastest growing population is a bit odd if then the focus is on this non-English speaking group.

AUTHORS: Thank you for this suggestion. We have added: "The literature has demonstrated several differences between Asians with and without lower English proficiency. In particular, compared to Asians with higher English proficiency, Asians with lower English proficiency have higher psychological distress (24,25), higher unmet healthcare needs (26), poorer quality of life and health (27,28), more limited access to care (26) and are less likely to adhere to screening guidelines (29) or receive and use health services (25,30)."

d. And then the focus become Vietnamese. This makes sense given that they have lower English proficiency, lower income and education, etc. But the authors link this to barriers to health service utilization. The connection to health services is not made clear. That should be fleshed out much more, that the primary concern here is health service utilization.

AUTHORS: We have added: " Indeed, a body of literature has documented lower utilization of various health services among U.S. Vietnamese when compared to other major Asian subgroups. For example, compared to Chinese and Korean populations in the U.S., fewer U.S. Vietnamese have a personal doctor as a main healthcare provider (31), have ever been tested for Hepatitis B (32), or have ever had colorectal screening (33). Moreover, a higher proportion of U.S. Vietnamese women have never had a Pap smear compared to Chinese and Cambodian women (34). Additionally, fewer U.S. Vietnamese women on average have ever sought mental health services compared to Chinese women or Filipino women in the U.S (35)."

e. Very little time in the intro is spent describing the HPV vaccine study. It would be good to flesh this out more too, to talk about how not having information from non-English speaking individuals could lead to incorrect results in a study like this.

AUTHORS: Thank you for this suggestion as well as the above request to bring in more information about the challenges in obtaining representative samples of U.S. Vietnamese. We have added: "U.S. Vietnamese have higher cervical cancer incidence rates than other racial/ethnic groups (36–39). A solution to reduce cervical cancer burden in this population is HPV vaccination. Unfortunately, our understanding of HPV vaccination among U.S. Vietnamese is limited, partly because large national probability surveys on HPV vaccine uptake in the U.S. typically do not supply separate statistics for U.S. Vietnamese. For example, the National Immunization Survey – Teen and the Behavioral Risk Factor Surveillance System, both of which provide estimates on HPV vaccine uptake, aggregate "Vietnamese" under the category of "Asians" and do not provide disaggregated Asian subgroup data in public-use datasets (40–42). The Health Information National Trends Survey (HINTS), which examines knowledge of the HPV vaccine, reports separate data for Vietnamese in some cycles but not all; the sample sizes for Vietnamese in the public-use datasets are also relatively small (~1%) which can hinder meaningful statistical modeling (43). Importantly, none of these surveys includes Vietnamese language versions of their questionnaires, which can be a major obstacle to research participation for U.S. Vietnamese with low English proficiency. In the context of research with HPV vaccination, this issue is particularly problematic, given that U.S. Vietnamese women with lower English proficiency also had lower HPV vaccine uptake (44). Consequently, excluding those with low English proficiency could bias estimates of HPV vaccine uptake and lead to an incomplete understanding of barriers underlying HPV vaccination in this population." 

2. Methods:

a. If only one parent per household was allowed to participate, would they have had the more English fluent person participate? Does this skew results?

AUTHORS: Thank you for this question. The survey was available in both English and Vietnamese. We have added more information about the translation process: "We used the Brislin’s back-translation method (45), an iterative translation process involving an independent translation of survey questions into Vietnamese and back-translation into English by two different translators, and then reviewed by the first author (who is fully fluent in both languages) and by approximately 10 Vietnamese native speakers to ensure comprehensibility."

3. Results:

a. It would be helpful to know how those who refused looked in terms of SES and other characteristics. Why did they refuse?

AUTHORS: Unfortunately, since SES and other characteristics were self-reported in the survey, we did not have these data from those who refused participation and therefore could not characterize this group. We also did not collect data on why eligible individuals may have refused to participate. We have added to the Limitations section: "We did not collect data on sociodemographic characteristics, acculturation, or mobile health usage of those who refused participation; therefore, we could not detect whether there were significant differences between those who chose to participate and those who did not. Additionally, we did not collect data on reasons why individuals refused participation."

b. The authors describe the differences in SES and in acculturation-related characteristics, but do not explore what this might mean to the results. A fuller analysis would strengthen the paper.

AUTHORS: We have added: "Researchers looking to replicate these recruitment methods for future research studies in Vietnamese populations should be mindful of subgroup variability in terms of sociodemographic characteristics and acculturation-related characteristics, even within a cultural population (e.g., U.S. Vietnamese). For health promotion efforts in particular, these differences point to the need to consider tailored approaches to cultural appropriateness (46). Knowing a person's educational background, level of Vietnamese cultural identity, degree of Vietnamese cultural knowledge, and ability to speak Vietnamese or English will facilitate crafting effective and resonating health messages (46). For example, in our sample, the majority of Facebook groups-recruited participants and those referred by them likely would require messages and programs in Vietnamese that leverage Vietnamese cultural information. In contrast, a considerable number of those recruited through CBOs and the first author's personal network may prefer information in Vietnamese that incorporates Vietnamese culture but may also be open to receiving information in English given their higher English proficiency. Moreover, findings from our study highlight the complexity of the relationships between participants' Vietnamese and American acculturation degrees, and consequently, the importance of including multidimensional acculturation-related measurements (47,48). For example, lower Vietnamese acculturation does not necessarily mean higher American acculturation (and vice-versa), as on average, those recruited through the first author's personal network had the highest scores for both Vietnamese and American cultural identity subscales."

c. In Table 3 - There is no description of the scale used for the acculturation measure, thus it is unclear what a score of 3.95 or 4.65 mean in relation to one another.

AUTHORS: We have added: "We also used the Asian American Multidimensional Acculturation Scale (15 items for each culture) (49) to separately assess cultural identity (6 items), cultural knowledge (3 items), language (4 items), and food consumption (2 items) for Vietnamese culture and American culture. Examples of questions included: "How much do you interact and associate with [Vietnamese people]/[typical American people]?"; "How much do you actually practice the traditions and keep the holidays of [Vietnamese culture]/[mainstream American culture]?"; "How well do you speak [Vietnamese]/[English]?"; and "How often do you actually eat [Vietnamese food]/[the food of mainstream American culture]?" Each item was scored on a 6-point Likert scale (0 – Not very much to 5 – Very much). A higher subscale score, derived as an average across subscale items, indicated higher cultural identity, cultural knowledge, language ability, or food consumption."

d. In Table 3 - Again here it would be helpful to know exactly how “Asian” is defined.

AUTHORS: We mentioned in the Methods section of the original submission: "We asked about participants' zip code and used data from the 2019 American Community Survey (19) to construct two variables capturing zip code-level percentage of Asians and percentage of Vietnamese." The zip code-level percentage of Asians is a variable listed in Table 3 and constructed from the 2019 American Community Survey data. 

e. In Table 3 and in text – consider separating out “mobile health usage” and “mobile phone usage.”

AUTHORS: We have separated this section per your suggestion into "daily use of mobile devices to access the Internet" and "past 12-month use of mobile health."

f. It is not clear where the results of the chi-square tests/tukey tests are reported in looking at differences across sociodemographic and other characteristics, or differences in pairs of means.

AUTHORS: Thank you for this question. As we described in the original submission, we used the chi-square tests and one-way analyses of variance to examine significant differences across sociodemographic characteristics, acculturation-related characteristics, and mobile health usage among recruitment methods. For any variables significant at the 0.05 level (described in the text of the Results section and shown in the p value columns of Table 2 and 3), we then conducted post-hoc analyses (the Tukey test for differences in pairs of means and a Tukey-type procedure for differences in pairs of proportions). Key findings from the post-hoc analyses are discussed in the text of the Results section, beginning with "Post-hoc analyses show that..."

g. Also, it would be great to have more details from the qualitative study. I believe there is only a single quote in the paper, but more details from the qualitative work could make the paper

AUTHORS: Thank you for this suggestion. We have added: "Another echoed the same sentiment: "Before filling out your survey, I didn’t know that boys should get the HPV vaccine too." In addition, a participant said: "This project is useful for me as it has helped me pay more attention to cervical cancer. I have heard about cervical cancer, but I have not paid much attention to the vaccine."

REFERENCES

1. Redmond N, Harker L, Bamps Y, Flemming SSC, Perryman JP, Thompson NJ, et al. Implementation of a Web-Based Organ Donation Educational Intervention: Development and Use of a Refined Process Evaluation Model. J Med Internet Res [Internet]. 2017 Nov 30;19(11):e396. Available from: http://www.jmir.org/2017/11/e396/

2. Rodriguez MD, Rodriguez J, Davis M. Recruitment of First-Generation Latinos in a Rural Community: The Essential Nature of Personal Contact. Fam Process [Internet]. 2006 Mar;45(1):87–100. Available from: http://doi.wiley.com/10.1111/j.1545-5300.2006.00082.x

3. Browne K. Snowball sampling: using social networks to research non‐heterosexual women. Int J Soc Res Methodol [Internet]. 2005 Feb;8(1):47–60. Available from: http://www.tandfonline.com/doi/abs/10.1080/1364557032000081663

4. Tung W-C, Nguyen DHT, Tran DN. Applying the transtheoretical model to cervical cancer screening in Vietnamese-American women. Int Nurs Rev [Internet]. 2008 Mar;55(1):73–80. Available from: http://doi.wiley.com/10.1111/j.1466-7657.2007.00602.x

5. Block P, Grund T. Multidimensional homophily in friendship networks. Netw Sci [Internet]. 2014 Aug 3;2(2):189–212. Available from: https://www.cambridge.org/core/product/identifier/S2050124214000174/type/journal_article

6. Lazarsfeld P, Merton RK. Friendship as a Social Process: A Substantive and Methodological Analysis. In: Berger M., Abel T., Charles H, editors. Freedom and Control in Modern Society. New York: Van Nostrand; 1954. p. 18–66. 

7. Magaço A, Munguambe K, Nhacolo A, Ambrósio C, Nhacolo F, Cossa S, et al. Challenges and needs for social behavioural research and community engagement activities during the COVID-19 pandemic in rural Mozambique. Glob Public Health [Internet]. 2021 Jan 2;16(1):153–7. Available from: https://www.tandfonline.com/doi/full/10.1080/17441692.2020.1839933

8. Archer-Kuhn B, Beltrano NR, Hughes J, Saini M, Tam D. Recruitment in response to a pandemic: pivoting a community-based recruitment strategy to facebook for hard-to-reach populations during COVID-19. Int J Soc Res Methodol [Internet]. 2021 Jun 16;1–12. Available from: https://www.tandfonline.com/doi/full/10.1080/13645579.2021.1941647

9. Health Management Associates. Community Based Organizations and COVID-19 [Internet]. [cited 2021 Jun 30]. Available from: https://www.healthmanagement.com/services/covid-19-resources-support/community-based-organization-needs/

10. Grantz KH, Meredith HR, Cummings DAT, Metcalf CJE, Grenfell BT, Giles JR, et al. The use of mobile phone data to inform analysis of COVID-19 pandemic epidemiology. Nat Commun [Internet]. 2020 Dec 30;11(1):4961. Available from: http://www.nature.com/articles/s41467-020-18190-5

11. Douedari Y, Alhaffar M, Duclos D, Al-Twaish M, Jabbour S, Howard N. ‘We need someone to deliver our voices’: reflections from conducting remote qualitative research in Syria. Confl Health [Internet]. 2021 Dec 17;15(1):28. Available from: https://conflictandhealth.biomedcentral.com/articles/10.1186/s13031-021-00361-w

12. Rahman SA, Tuckerman L, Vorley T, Gherhes C. Resilient Research in the Field: Insights and Lessons From Adapting Qualitative Research Projects During the COVID-19 Pandemic. Int J Qual Methods [Internet]. 2021 Jan 1;20:160940692110161. Available from: http://journals.sagepub.com/doi/10.1177/16094069211016106

13. Brody AA, Convery KA, Kline DM, Fink RM, Fischer SM. Transitioning to Remote Recruitment and Intervention: A Tale of Two Palliative Care Research Studies Enrolling Underserved Populations during COVID-19. J Pain Symptom Manage [Internet]. 2021 Jun; Available from: https://linkinghub.elsevier.com/retrieve/pii/S0885392421004000

14. Melis G, Sala E, Zaccaria D. Remote recruiting and video-interviewing older people: a research note on a qualitative case study carried out in the first Covid-19 Red Zone in Europe. Int J Soc Res Methodol [Internet]. 2021 Apr 15;1–7. Available from: https://www.tandfonline.com/doi/full/10.1080/13645579.2021.1913921

15. Hensen B, Mackworth-Young CRS, Simwinga M, Abdelmagid N, Banda J, Mavodza C, et al. Remote data collection for public health research in a COVID-19 era: ethical implications, challenges and opportunities. Health Policy Plan [Internet]. 2021 Apr 21;36(3):360–8. Available from: https://academic.oup.com/heapol/article/36/3/360/6130108

16. Sadler GR, Lee H-C, Lim RS-H, Fullerton J. Research Article: Recruitment of hard-to-reach population subgroups via adaptations of the snowball sampling strategy. Nurs Health Sci [Internet]. 2010 Sep;12(3):369–74. Available from: http://doi.wiley.com/10.1111/j.1442-2018.2010.00541.x

17. Salganik MJ, Heckathorn DD. 5. Sampling and Estimation in Hidden Populations Using Respondent-Driven Sampling. Sociol Methodol [Internet]. 2004 Dec 24;34(1):193–240. Available from: http://journals.sagepub.com/doi/10.1111/j.0081-1750.2004.00152.x

18. Heckathorn DD. Comment: Snowball versus Respondent-Driven Sampling. Sociol Methodol [Internet]. 2011 Aug 19;41(1):355–66. Available from: http://journals.sagepub.com/doi/10.1111/j.1467-9531.2011.01244.x

19. United States Census Bureau. 2019 American Community Survey 5-Year Estimates Data Profiles [Internet]. 2020. Available from: http://data.census.gov

20. Holland AT, Palaniappan LP. Problems With the Collection and Interpretation of Asian-American Health Data: Omission, Aggregation, and Extrapolation. Ann Epidemiol. 2012 Jun;22(6):397–405. 

21. Srinivasan S, Guillermo T. Toward improved health: disaggregating Asian American and Native Hawaiian/Pacific Islander data. Am J Public Health [Internet]. 2000 Nov;90(11):1731–4. Available from: http://www.ncbi.nlm.nih.gov/pubmed/11076241

22. Islam NS, Khan S, Kwon S, Jang D, Ro M, Trinh-Shevrin C. Methodological Issues in the Collection, Analysis, and Reporting of Granular Data in Asian American Populations: Historical Challenges and Potential Solutions. J Health Care Poor Underserved. 2010;21(4):1354–81. 

23. United States Census Bureau. 2019 American Community Survey 1-Year Estimates Total Population [Internet]. 2019 [cited 2020 Sep 27]. Available from: http://data.census.gov

24. Zhang W, Hong S, Takeuchi DT, Mossakowski KN. Limited English proficiency and psychological distress among Latinos and Asian Americans. Soc Sci Med [Internet]. 2012 Sep;75(6):1006–14. Available from: https://linkinghub.elsevier.com/retrieve/pii/S027795361200425X

25. Kim G, Worley CB, Allen RS, Vinson L, Crowther MR, Parmelee P, et al. Vulnerability of Older Latino and Asian Immigrants with Limited English Proficiency. J Am Geriatr Soc [Internet]. 2011 Jul;59(7):1246–52. Available from: http://doi.wiley.com/10.1111/j.1532-5415.2011.03483.x

26. Jang Y, Kim MT. Limited English Proficiency and Health Service Use in Asian Americans. J Immigr Minor Heal [Internet]. 2019 Apr 24;21(2):264–70. Available from: http://link.springer.com/10.1007/s10903-018-0763-0

27. Jang Y, Yoon H, Park NS, Chiriboga DA. Health Vulnerability of Immigrants with Limited English Proficiency: A Study of Older Korean Americans. J Am Geriatr Soc [Internet]. 2016 Jul;64(7):1498–502. Available from: http://doi.wiley.com/10.1111/jgs.14199

28. Gee GC, Ponce N. Associations Between Racial Discrimination, Limited English Proficiency, and Health-Related Quality of Life Among 6 Asian Ethnic Groups in California. Am J Public Health [Internet]. 2010 May;100(5):888–95. Available from: http://ajph.aphapublications.org/doi/10.2105/AJPH.2009.178012

29. Sentell T, Braun KL, Davis J, Davis T. Colorectal Cancer Screening: Low Health Literacy and Limited English Proficiency Among Asians and Whites in California. J Health Commun [Internet]. 2013 Dec 4;18(sup1):242–55. Available from: http://www.tandfonline.com/doi/abs/10.1080/10810730.2013.825669

30. Sentell T, Shumway M, Snowden L. Access to Mental Health Treatment by English Language Proficiency and Race/Ethnicity. J Gen Intern Med [Internet]. 2007 Nov 24;22(S2):289–93. Available from: http://link.springer.com/10.1007/s11606-007-0345-7

31. Tran H, Do V, Baccaglini L. Health Care Access, Utilization, and Management in Adult Chinese, Koreans, and Vietnamese with Cardiovascular Disease and Hypertension. J Racial Ethn Heal Disparities [Internet]. 2016 Jun 28;3(2):340–8. Available from: http://link.springer.com/10.1007/s40615-015-0155-2

32. Ma GX, Tan Y, Wang MQ, Yuan Y, Chae WG. Hepatitis B Screening Compliance and Non-compliance among Chinese, Koreans, Vietnamese and Cambodians. Clin Med Gastroenterol [Internet]. 2010 Jan 5;3:CGast.S3732. Available from: http://journals.sagepub.com/doi/10.4137/CGast.S3732

33. Ma GX, Wang MQ, Toubbeh J, Tan Y, Shive S, Wu D. Factors Associated with Colorectal Cancer Screening Among Cambodians, Vietnamese, Koreans and Chinese Living in the United States. N Am J Med Sci (Boston) [Internet]. 2012 Jan;5(1):1–8. Available from: http://www.ncbi.nlm.nih.gov/pubmed/23243486

34. Ma GX, Toubbeh JI, Wang MQ, Shive SE, Cooper L, Pham A. Factors Associated With Cervical Cancer Screening Compliance and Noncompliance among Chinese, Korean, Vietnamese, and Cambodian Women. J Natl Med Assoc [Internet]. 2009 Jun;101(6):541–51. Available from: https://linkinghub.elsevier.com/retrieve/pii/S0027968415309391

35. Appel HB, Huang B, Ai AL, Lin CJ. Physical, Behavioral, and Mental Health Issues in Asian American Women: Results from the National Latino Asian American Study. J Women’s Heal [Internet]. 2011 Nov;20(11):1703–11. Available from: http://www.liebertpub.com/doi/10.1089/jwh.2010.2726

36. Parker SL, Davis KJ, Wingo PA, Ries LA, Heath CW. Cancer statistics by race and ethnicity. CA Cancer J Clin [Internet]. 1998 Jan 1;48(1):31–48. Available from: http://doi.wiley.com/10.3322/canjclin.48.1.31

37. Miller BA, Kolonel LN, Bernstein L, Young JL, Swanson GM, West D, et al. Racial/Ethnic Patterns of Cancer in the United States 1988-1992. NIH Pub. N. Miller BA, editor. Bethesda, MD: National Cancer Institute; 1996. 

38. Surveillance Epidemiology and End Results Program. Cancer Stat Facts: Cervical Cancer [Internet]. Available from: https://seer.cancer.gov/statfacts/html/cervix.html

39. Jin H, Pinheiro PS, Xu J, Amei A. Cancer incidence among Asian American populations in the United States, 2009-2011. Int J Cancer [Internet]. 2016 May 1;138(9):2136–45. Available from: http://doi.wiley.com/10.1002/ijc.29958

40. NORC at the University of Chicago. National Immunization Survey - Teen. A Codebook for the 2019 Public-Use Data File [Internet]. 2020. Available from: https://www.cdc.gov/vaccines/imz-managers/nis/downloads/NIS-TEEN-PUF19-CODEBOOK.pdf

41. Elam-Evans LD, Yankey D, Singleton JA, Sterrett N, Markowitz LE, Williams CL, et al. National, Regional, State, and Selected Local Area Vaccination Coverage Among Adolescents Aged 13–17 Years — United States, 2019. MMWR Morb Mortal Wkly Rep [Internet]. 2020 Aug 21;69(33):1109–16. Available from: http://www.cdc.gov/mmwr/volumes/69/wr/mm6933a1.htm?s_cid=mm6933a1_w

42. Centers for Disease Control and Prevention. LLCP 2019 Codebook Report. Overall version data weighted with _LLCPWT. Behavioral Risk Factor Surveillance System. July 31, 2020 [Internet]. 2020 [cited 2021 Jul 6]. Available from: https://www.cdc.gov/brfss/annual_data/2019/pdf/codebook19_llcp-v2-508.HTML

43. National Cancer Institute. Public Use Dataset (Health Information National Trends Survey) [Internet]. [cited 2021 Jul 6]. Available from: https://hints.cancer.gov/data/download-data.aspx

44. Yi JK, Anderson KO, Le YC, Escobar-Chaves SL, Reyes-Gibby CC. English proficiency, knowledge, and receipt of HPV vaccine in Vietnamese-American Women. J Community Health. 2013;38(5):805–11. 

45. Brislin RW. Back-Translation for Cross-Cultural Research. J Cross Cult Psychol. 1970 Sep 1;1(3):185–216. 

46. Kreuter MW, Lukwago SN, Bucholtz DC, Clark EM, Sanders-Thompson V. Achieving Cultural Appropriateness in Health Promotion Programs: Targeted and Tailored Approaches. Heal Educ Behav [Internet]. 2003 Apr 1;30(2):133–46. Available from: http://journals.sagepub.com/doi/10.1177/1090198102251021

47. Berry JW. Theories and Models of Acculturation [Internet]. Schwartz SJ, Unger J, editors. Vol. 1. Oxford University Press; 2017. Available from: http://oxfordhandbooks.com/view/10.1093/oxfordhb/9780190215217.001.0001/oxfordhb-9780190215217-e-2

48. Sam DL, Berry JW. Acculturation: When Individuals and Groups of Different Cultural Backgrounds Meet. Perspect Psychol Sci [Internet]. 2010;5(4):472–81. Available from: http://www.jstor.org/stable/41613454

49. Gim Chung RH, Kim BSK, Abreu JM. Asian American Multidimensional Acculturation Scale: Development, Factor Analysis, Reliability, and Validity. Cult Divers Ethn Minor Psychol [Internet]. 2004;10(1):66–80. Available from: http://www.ncbi.nlm.nih.gov/pubmed/14992631

---

## [Editor Report · Decision Letter 1]

2 Aug 2021

Experience and lessons learned from multi-modal internet-based recruitment of U.S. Vietnamese into research

PONE-D-21-14089R1

Dear Author,

We’re pleased to inform you that your manuscript has been judged scientifically suitable for publication and will be formally accepted for publication once it meets all outstanding technical requirements.

Kind regards,

Marcel Pikhart

Academic Editor

PLOS ONE
---

## [Editor Report · Acceptance letter]

6 Aug 2021

PONE-D-21-14089R1 

Experience and lessons learned from multi-modal internet-based recruitment of U.S. Vietnamese into research 

Dear Dr. Vu:

I'm pleased to inform you that your manuscript has been deemed suitable for publication in PLOS ONE. Congratulations! Your manuscript is now with our production department. 

Kind regards, 

on behalf of

Dr. Marcel Pikhart 

Academic Editor

PLOS ONE